# Pellets Inoculated with *Bacillus amyloliquefaciens* H57 Modulates Diet Preference and Rumen Factors Associated with Appetite Regulation in Steers

**DOI:** 10.3390/ani11123455

**Published:** 2021-12-04

**Authors:** Thi Thuy Ngo, Nguyen N. Bang, Peter Dart, Matthew Callaghan, Athol Klieve, David McNeill

**Affiliations:** 1School of Veterinary Science, The University of Queensland, Gatton, QLD 4343, Australia; nn.bang@uq.net.au (N.N.B.); d.mcneill@uq.edu.au (D.M.); 2Faculty of Animal Science, Vietnam National University of Agriculture, Hanoi 131000, Vietnam; 3School of Agriculture and Food Sciences, The University of Queensland, Gatton, QLD 4343, Australia; p.dart@uq.edu.au; 4Ridley AgriProducts Pty Ltd., Toowong, QLD 4066, Australia; Matthew.Callaghan@ridley.com.au; 5Queensland Alliance for Agriculture and Food Innovation, The University of Queensland, St Lucia, QLD 4069, Australia; a.klieve@uq.edu.au

**Keywords:** *Bacillus amyloliquefaciens* H57, feed preference, ruminal pH, ruminal VFA, appetite

## Abstract

**Simple Summary:**

The probiotic *Bacillus amyloliquefaciens* strain H57 (H57) may reinforce preferential feeding behaviour by changing ruminal fermentation parameters. Four rumen-fistulated steers were offered feedlot pellets, with (H57) or without (Control, C) the H57 probiotic. Half of the pellets were added to the rumen, at time zero, and half were offered for oral consumption over the next six hours, to make four feeding treatments. Each steer was offered each treatment over time. Each offering was over six days, with rumen fluid sampled over the last three days for a six-hour period per day. A five-minute preference test was performed at the end of each rumen sampling period by simultaneously offering the steers 4 kg of H57 and C pellets. The steers preferred the H57 over the C pellets but the route of offering (rumen versus oral) had no effect on preference. Ruminal pH and molar proportions of iso-butyrate and iso-valerate were higher and ammonia concentrations tended to be greater for H57 compared to C. However, since the route of offering had no effect on preference, the hypothesis, that ruminal fermentation changes take precedence over oral (taste) sensations in driving preference, was not supported.

**Abstract:**

This study examined whether the probiotic *Bacillus amyloliquefaciens* strain H57 (H57) affects ruminal fermentation parameters that exercise post-ingestive feedback appetite control mechanisms. A 4 × 4 Latin square design was used to separate pre- and post-ingestive effects of H57 in four rumen-fistulated steers. The steers were offered a set amount of feedlot pellets, inoculated with H57 or without H57 (control, C). Half of the total amount of pellets fed were introduced intra-ruminally (r), and then the remaining pellets were orally consumed (o) to make four feeding treatments: H57r/H57o, H57r/Co, Cr/H57o and Cr/Co. Rumen fluid was sampled at 2, 4 and 6 h after feeding. Preference behaviour was tested immediately after the 6 h rumen fluid sampling by simultaneously offering the steers 4 kg of each of H57 and C pellets in adjacent troughs for 5 min. Steers preferred the pellets with added H57 over the C pellets (56:44; *p* < 0.001) and their preferences were not affected by the treatment protocol imposed to separate post- from pre-ingestive effects (*p* > 0.05). Steers fed H57 pellets had higher ruminal pH, molar proportions of iso-butyrate and iso-valerate (*p* < 0.05) and tended to have greater ruminal ammonia concentrations compared to those fed C pellets (*p* < 0.1). However, post-ingestive signals did not affect diet preference more than pre-ingestive signals.

## 1. Introduction

Reduced appetite and feed intake are common clinical signs of stress in cattle [1] and the amelioration thereof is increasingly a regulated requirement in the ruminant industries [2]. Probiotics can improve feed intake in ruminants, thus sustaining their performance and wellbeing during stressful situations as reviewed by [3,4,5,6]. Ramsing [7] and Yuan [8] suggested that supplementing fermented yeast culture obtained from *Saccharomyces cerevisiae* during the transition period increased the number of meals consumed by pre-partum cows, which ultimately led to an improvement in DMI. Likewise, the length of the daily meals and feeding times were greater for postpartum cows supplemented with fermented yeast culture compared with those that were not [7]. Diets containing the spores of *Bacillus licheniformis* and *B. subtilis* have stimulated the starter mixture intake of milk-fed calves [9]. Recently, H57 was shown to enhance the DMI of pregnant ewes and dairy calves fed pellets pre-inoculated with the H57 [10,11].

H57 is a spore-forming bacterium that was isolated from lucerne leaves and developed as an inoculant in the hay-making process to reduce the risk of fungal spoilage [12]. When tested to indicate if ruminants would accept grass/clover hay containing H57, mature ewes preferred the H57-treated hay over the untreated hays [12]. Le [13] subsequently found the preference proportion of approximately 70:30 in weaned dairy calves for H57-inoculated calf pellets over un-inoculated pellets. This response was recorded within 6 h on the first day of offering the inoculated pellets after several days of ensuring that the calves had adjusted to the testing procedure. The 70:30 ratio remained the same for three consecutive days and on the fourth day, despite attempts to stimulate the calves’ preference for the un-inoculated pellets by adding a glucose sweetener, the preference for H57 pellets persisted. However, the mechanisms explaining the link between H57 supplementation, increased diet preference and intake of ruminants remain unclear.

Diet preference and feed intake of ruminants are controlled by neurally mediated interactions between feed sensory characteristics and post-ingestive feedback [14]. Animals experienced sensory cues before the feed was swallowed and post-ingestive feedback, which includes digestive and metabolic signals experienced by the animals, after the feed was swallowed [15]. Favreau [16] and Baumont [17] highlighted how the sensory characteristics of a feed were used as a tool by ruminants to identify and choose between feeds. For example, sheep could use artificial flavours in feed as indicators to direct their preference and intake over relatively short-term periods [18]. In diet preference studies, post-ingestive feedback was considered predominant in the determination of feed intake in ruminants [14,15,19]. The post-ingestive feedback could include ruminal fermentation parameters such as pH, concentrations of VFAs, and its profile or NH_3_ concentrations [20]. Forbes [21] showed the connection between low ruminal pH and the regulation of diet preference as ruminal pH sent a feedback signal to the central nervous system to depress appetite and feed intake. Similarly, a high concentration of individual VFA, particularly propionate, can alter diet selection [21]. An elevation in the ruminal concentration of NH_3_ can also induce satiety and thus reduce feed intake [22,23].

Despite the various aspects of feed intake regulation that have been elucidated in ruminants, we are not aware of any reports on how bacterial probiotics might regulate diet preference. There are indications that the addition of probiotics to diets can alter ruminal pH and concentrations of fermentation products (e.g., ammonia and total and individual VFAs) and that such changes may be associated with appetite regulation [10,11,24,25,26]. Thus, this study tested the hypothesis that H57 could improve diet preference in steers through changes in rumen fermentation parameters related to diet preference and feed intake regulation.

## 2. Materials and Methods

### 2.1. Animals and Experimental Design

Four mature *Bos indicus* cross, rumen cannulated, steers (mean liveweight (LW): 900 ± 37 kg, age 5 years old) were used. They were housed in individual pens (3 × 10 m) with rubberised flooring at the Queensland Animal Science Precinct, the University of Queensland (UQ) (Gatton, QLD, Australia). All experimental procedures involving animals were approved by the Animal Ethics Committee of the UQ (SVS/451/16).

Throughout the experiment, a restricted amount of beef feedlot pellets inoculated with (H57) or without (Control, C) H57 were offered at 9:00 a.m. and at 3:15 p.m. Outside those times, chaffed Rhodes grass hay and water were provided ad libitum. The experiment was designed to dissociate the effects of pre-ingestive characteristics and post-ingestive feedback on diet preference in steers, based on a procedure similar to that used by Favreau [15]. The experiment was conducted over 64 days with four 16-day periods and five days washout between each period to minimise any carryover effects to the next period. The steers received either H57 or C pellets placed directly into the rumen (r), followed by either type of pellet presented in a trough so they could consume normally (orally, o). The four treatments were H57r/H57o, H57r/Co, Cr/H57o and Cr/Co. Treatments were allocated to the four steers in a complete Latin square design where each steer would receive each of the four treatments over the four treatment phases of 6 days with each treatment phase preceded by an adjustment phase of 10 days (Figure 1).

On days 1–3 of each 10-day adjustment phase, about 1 kg of a 50:50 mix of H57 and C (500 g, as fed of each) was offered to familiarise the steers with both pellet types and reduce carryover effects between treatments. From day 4 to 10, for maintenance of LW, all steers were given a 50:50 mix of H57 and C pellet amounts equivalent to 0.35% DM of steer LW. For the subsequent 6-day treatment phase, the steers continued to be fed orally at the same rate of the pellet type they had been given in the previous 7-day (A4–A10) portion of the adjustment period. Additionally, at 8:00 a.m., either H57 or C pellets were introduced into the rumen via the fistula as per the experiment design at 50% of the weight of pellets that the steers consumed in the preceding A4 to A10 adjustment phase. These pellets were mixed 1:1 with artificial saliva (Kansas State Buffer) [27] immediately prior to introduction into the rumen. The remaining half of the total treatment pellet type was presented for oral consumption at 9:00 a.m. for 15 min and then chaffed Rhodes grass hay was provided ad libitum. Immediately after the 3:00 p.m. rumen sampling, a preference test was conducted to determine which type of pellet the steer preferred to assess the influence of the preceding treatment at 8:00 a.m. (feeding directly into rumen) and 9:00 a.m. (trough feeding). At the 3:15 p.m. test feeding, 4 kg (as fed) of each of the pellets was offered simultaneously in the two adjacent troughs, one containing H57 pellets and the other C pellets, and steers were allowed to eat freely from each for 5 min. Each day the positions of the troughs were reversed to prevent any bias. The relative preference for H57 pellets was calculated as (intake of H57 pellets)/ (intake of C plus H57 pellets) × 100 [28].

### 2.2. Preparation of Beef Feedlot Pellets

The method used to prepare H57 inoculum for the cattle trial was as described by Schofield [29]. Briefly, the production of the H57 inoculum was performed in a series of batch cultures, then cultivated in a 100 L fermenter for the production of spores. The fermenter contents were centrifuged at 15,000 rpm to harvest the bacteria and suspended material from the fermenter. The paste from centrifuging the fermenter contents was mixed with sodium bentonite and freeze-dried. The product was then ground into a powder for subsequent use. The H57 bentonite powder (6.4 × 10^10^ cfu spores/g) was provided to inoculate 3 tons of feed at 1 × 10^6^ cfu spores/g pellet. The bentonite powder was mixed with ground wheat in a feed mixer, and this diluted inoculum was incorporated into a final premix for steam pelleting in the Ridley Agriproducts Pty Ltd. feedmill at Toowoomba, QLD, Australia. The pellets were cooled and stored in 20 kg plastic bags. The bagged pellets were stored at 7 °C for 3 months and sufficient bags were removed to ambient conditions at the beginning of each period for feeding. The ingredients and chemical composition of the pellets are presented in Table 1.

### 2.3. Sample Collection and Measurements

Approximately 0.5 kg subsamples of feed offered to steers and refusals were collected daily and stored in a cool room at 7 °C. At the end of the experiment, these daily samples were pooled, mixed completely, then about 0.5 kg of hay and 1 kg of pellet subsamples were stored at −20 °C for chemical analysis. Samples of 100 g for the plate counts of the H57 populations were stored at 4 °C.

Rumen contents were sampled at 8:00 a.m. (0 h) before introducing the pellets into the rumen and then at 11:00 a.m. (2 h), 1:00 p.m. (4 h) and 3:00 p.m. (6 h) on the last 3 days of the experimental periods. Rumen samples were collected through the fistula by first sliding into the rumen a PVC tube (40 mm ID; 60 cm length; Holman Industries HQ, Osborne Park, WA, Australia), with nylon stockings (Bonds, Melbourne, VIC, Australia) glued over the tube end inserted into the rumen to strain the rumen fluid. Then a smaller tube (10 mm ID; 80 cm length; Holman Industries HQ, Osborne Park, WA, Australia) was inserted inside the larger tube, which was attached to a 60 mL syringe (Terumo, Somerset, NJ, USA) that was used to withdraw about 40 mL of the rumen fluid. The rumen fluid was then placed into sterile 250 mL plastic containers (Thermo Fisher Scientific, Waltham, MA, USA). The ruminal fluid pH was measured within minutes of collection using a portable pH meter (Eutech pH 6^+^, Eutech Instruments, Ayer Rajah Crescent, Singapore). Two subsamples were transferred into 10 mL tubes (Thermo Fisher Scientific, Waltham, MA, USA) and placed in an ice bath. A 4 mL sample of the rumen fluid was preserved by adding 1 mL of 20% metaphosphoric acid for the analysis of total VFA and its profile. A separate 8 mL rumen filtrate was preserved by adding 2 mL of 0.5 M sulphuric acid for analysis of ammonia content. These rumen subsamples were stored at −20 °C for subsequent analysis.

### 2.4. Cell Counting Procedure for H57 Spore and Vegetative

The viable count method of Harrigan and McCance [30] was modified and used to count H57 concentrations (spore and vegetative cells) in the pellet samples. The procedure is described in detail in Ridley-ARC [31]. Briefly, 1.0 g of powdered material was weighed into a sterilized 200 mL beaker and 100 mL sterilized chilled water added. The suspension was then mixed for 2 min at 24,000 rpm using a T25 digital Ultra-Turrax IKA homogenizer with a 25 mm dispersing tool (IKA, Staufen, Germany). Three independent 0.1 mL aliquots were taken from the feed suspension and mixed with 0.9 mL sterile water in sterile 1.5 mL Eppendorf tubes. The tubes were labelled as A, B and C samples. To count total cells, 0.1 mL aliquots (one replicate each from A, B and C tubes) was spread on nutrient agar and labelled appropriately. The Eppendorf tubes were then heated for 20 min at 80 °C in a heating block (1572VWR, VWR^TM^, Radnor, PA, USA) to kill vegetative cells, and a repeat 0.1 mL aliquot spread onto nutrient agar to count spores. Cells were grown overnight at 28–30 °C and the colonies on the plate were tabulated. Vegetative cells were determined as total cells minus spores. The C pellets had no H57 while the H57 pellets had cell counts that ranged from 0.6 to 0.9 × 10^6^ cfu spores/g pellets).

### 2.5. Chemical Analysis

The feed samples were analysed for nutritive contents by Dairy One Forage Laboratory (Ithaca, NY, USA). All samples were analysed using the wet chemistry procedures for dry matter (DM) (method 930.15), crude protein (CP) (method 984.13), fat (method 920.39), minerals (method 985.01), starch (method 996.11) [32], acid detergent fibre (ADF) and neutral detergent fibre (NDF) [33].

The frozen rumen fluid samples were thawed completely and then centrifuged at 2000× *g* for 20 min (Eppendorf^®^ Minispin^®^, Hamburg, Germany) to separate the liquid phase. The clear supernatants were collected for VFA analysis. Separation of VFAs was determined by the gas-liquid chromatography method (Cottyn and Boucque, 1968) and performed on a GC-2010 plus (Shimadzu, Kyoto, Japan) using a polar capillary column (ZB-FFAP 30 m × 0.53 mm × 1.0 µm, Zebron^®^, Phenomenex, Inc., Torrance, CA, USA). The temperature program of the GC was set initially at 85 °C for four minutes then increased to 200 °C at a rate of 15 °C/min. The flow rate of carrier gas (high purity helium) was at 5.0 mL/min, 67 kPa for two minutes then from 1.8 kPa/min to 81 kPa. Individual VFAs were quantified at 210 °C by a Chromatopac C-R6A (Shimadzu, Kyoto, Japan) equipped with a flame-ionisation detector. Ammonia concentrations were determined by the distillation method [32] using the Buchi 321 distillation unit (Flawil, St Gallen, Switzerland) and then titrated against 0.01M HCl using the automatic titration unit (TitraLab 845, Hach, Loveland, CO, USA). The operating procedure followed the guidelines recommended by the manufacturers.

### 2.6. Statistical Analysis

Data were analysed using the mixed model [34] of the statistical software package SAS, Version 9.4 (AS Institute Inc., Cary, NC, USA) [35]. The results were presented as least-squares mean ± s.e.m. The differences between treatment effects were assessed by Tukey multiple comparisons. The differences were considered significant at *p* < 0.05 and statistical tendency was declared at *p* < 0.10.

The following statistical models were adopted:Y_ijk_ = μ + Treatment_i_ + Steer_j_ + Period_k_ + Treatment_i_ Period_k_ + e_ijk_
Y_ijkl_ = μ + Oral_i_ + Rumen_j_ + Steer_k_ + Period_l_ + Oral_i_ Rumen_j_ + Oral_i_ Rumen_j_ Period_l_ + e_ijkl_
where Y_ijkl_ is the variable response; µ, overall mean; Treatment_i (i = 1 to 4)_, fixed effect of treatment (H57r/H57o vs. H57r/Co vs. Cr/ H57o vs. Cr/Co); Steer_j,k (j,k = 1 to 4)_, random effect of the individual steer within the square; Period_k,l (k,l = 1 to 4)_, fixed effect of the period within the square; Treatment_i_ Period_k_, fixed effect of the interaction between treatment and period; Oral_i (i= 1 to 2)_, fixed effect of orally consuming H57 (H57r/H57o and H57r/H57o) vs. C pellets (H57r/Co and Cr/Co); Rumen_j (j = 1 to 2)_, fixed effect of intra-ruminally consuming H57 (H57r/H57o and H57r/Co) vs. C pellets (Cr/H57o and Cr/Co); Oral_i_ Rumen_j_, fixed effect of the interaction between orally and intra-ruminally consuming pellet treatments; Oral_i_ Rumen_j_ Period_k_, fixed effect of the interaction among orally and intra-ruminally consuming pellet treatments and period; e_ijk_ and e_ijkl,_ are residual errors. All interactions were systematically removed from the model when they were non-significant, and a reduced model was used to determine treatment effects.

A one-sample *t*-test was used to compare actual preference data, with 0.5 as the reference value. Data for ruminal pH, NH_3_ and total and individual VFA levels were analysed by sampling time using a repeated measure with compound symmetry.

## 3. Results

### 3.1. Daily Feed Intake and Short-Term Preference Tests

The route of pellets addition into the digestive system did not affect pellet DMI in a five-minute preference test (*p* = 0.87; Table 2). Over 24 h, the total pellet intake of the steers was similar between treatments (*p* = 0.90). Likewise, differences in total hay intake due to treatments were not detected (*p* = 0.52).

Preference was not influenced by whether the pellet type had been previously introduced orally or intra-ruminally (*p* > 0.05; Figure 2). However, when considered separately from the morning feeding treatments, the steers’ preference for H57-inoculated pellets during the short-term preference test was consistent across treatments (*p* < 0.05). When the data were combined across all pellet treatments, the average preference of animals for the H57 pellets compared with their preference for the C pellets was approximately 56:44 (*p* < 0.001).

### 3.2. Ruminal pH

Across the treatments, there was no difference in the pattern of ruminal pH during the 6 h period after feeding the pellets (*p* > 0.05; Figure 3). Ruminal pH declined to reach a minimum at 4 h post-feeding and began to recover towards the 6 h point but did not reach pre-feeding values (0 h) by that time. When the pellets were introduced directly into the rumen via the rumen fistula, ruminal pH at 4 h was higher for the H57 compared with the C pellets (6.20 vs. 6.12; *p* = 0.04) with a trend towards higher rumen pH appearing by 2 h (6.38 vs. 6.32; *p* = 0.08) and extending to 6 h (6.35 vs. 6.30; *p* = 0.06).

### 3.3. Total Concentration and Molar Proportion of Individual VFAs in the Rumen

There were no differences detected in the total VFA concentrations across the pellet treatments (*p* > 0.05; Figure 4). The total VFA concentrations increased to achieve the highest level at 4 h after the morning feeding and then reduced towards the end of the 6 h time point.

The molar proportions of individual rumen VFAs including acetate, propionate, n-butyrate and n-valerate were generally similar among the pellet treatments (*p* > 0.05; Figure 5A–D). By contrast, at the 6 h time point, the molar proportions of iso-butyrate and iso-valerate were affected by the treatment (*p* < 0.05; Figure 5E,F). All treatments in which H57 was consumed orally or introduced directly into the rumen (H57r/H57o, Cr/H57o, H57r/Co), when averaged, had approximately 19% and 30% higher molar proportions of iso-butyrate and iso-valerate, respectively, than the Cr/Co treatment.

### 3.4. Ruminal Ammonia

No treatment effect was detected for ruminal NH_3_ concentration at 6 h (*p* > 0.05; Figure 6). However, at 2 h and 4 h, the NH_3_ concentrations for the treatment in which H57 pellets were consumed orally and introduced directly into the rumen (H57r/H57o) were 217.6 and 119 mg/L respectively, which tended to be greater than the remaining three treatments (*p* = 0.07 and *p* = 0.09, respectively).

## 4. Discussion

Improvements in post-ingestive signals, such as changes in ruminal pH, NH_3_ and VFA concentration and profiles consequential to the inoculation of feed with H57, were expected to drive the preference decisions of the steers. However, no such effect was detected. Even though the steers consistently preferred the H57-inoculated over the un-inoculated pellets, and ruminal pH, NH_3_ and iso-acids were elevated in the short term due to the H57, these responses were unrelated to the treatment protocol, which was imposed to separate post- from pre-ingestive (i.e., taste and odour) effects. Interpretations could be that the post-ingestive signals were insufficiently extreme to initiate a preference response, or that H57 improves feed preference through a combination of pre- and post-ingestive signals.

The protocol—imposed to separate the pre- and post-ingestive effects of a feed—was similar to that of Favreau [15] but with some differences. They used sheep, whereas this study used steers. They compared hays, whereas this study compared pellets. After a feed was given intra-ruminally (r) at 0 h, they offered hay ad libitum at trough (o) over several hours within a 6 h period, whereas this study offered a set amount of a given pellet after the ruminal introduction that was consumed within minutes. It was unable to offer the pellet ad libitum in the current study, as it would have put the steers at risk of ruminal acidosis and laminitis. Consequently, this study was unable to measure the rate of intake in the 6 h period. However, as did Favreau [15], this study applied one or the other feeds intra-ruminally at 0 h followed by one or the other feeds offered at trough and a five-minute preference test that was expected to be affected by the preceding four feeding treatment combinations. The logic was that there was a post-ingestive advantage afforded by H57 if ruminal parameters and preference for H57 (in the five-minute preference test) increased after H57r/H57o vs. Cr/H57o or H57r/Co vs. Cr/Co, whereas a pre-ingestive effect would have been determined if ruminal parameters and preference for H57 increased after Cr/H57o vs. H57r/Co. Since none of these comparisons were different, it is concluded that there was no evidence that H57 caused either a pre- or post-ingestive advantage.

On the five minute preference test, steers consistently preferred H57-inoculated over the un-inoculated pellets. This was not influenced by whether the pellets were introduced orally, indicating no short-term (i.e., within the 6 h testing period) overriding effect of pre- or post-ingestive feedback on diet preference. The preference for H57 pellets observed in this study was consistent with Le [13], who found that the weaned dairy calves demonstrated a 70% preference for H57 pellets. A lower preference for H57 pellets recorded in the current study compared with that of Le [13] could be explained by the differences in the ages of the animals and methods to assess diet preference. While Le [13] used weaned dairy calves at 20 weeks of age, this current study used mature steers that were 5 years of age. Additionally, Le [13] conducted a pellet preference test that lasted for 6 h, whereas the current study assessed the pellet preference of steers for only 5 min. Miller-Cushon [36] suggested that diet preference tests conducted or repeated over a long period might induce results that were different to that of short-term preference studies. Le [13] also concluded that the sweet taste was not a driver of preference for H57 pellets. It is speculated H57 enhanced diet preference may be related to the ability to prevent feed degradation. H57 was developed as an inoculant to protect stored forages and grain-based pellets against spoilage by fungi and other feed-spoiling organisms [12,37]. Further, Schofield [38] reported that H57 had the capacity to produce several kinds of lipopeptides and polyketide compounds, which are biological control agents and thought to inhibit fungal and bacterial growth. If the H57 spores germinate and grow in the inoculated pellets during storage, then the vegetative cells may subsequently produce antimicrobial compounds that are effective at controlling microbial contamination. This may preserve the freshness of H57-inoculated pellets, thereby influencing diet preference.

Ruminal pH was observed to be higher for the H57 than the C pellets at 4 h. Provenza [14] stated that the preference for a feed was adjusted according to its post-ingestive consequences, and on that basis, animals remembered to select or avoid that feed. This feedback could include ruminal pH, which is highly related to intake rate [39]. In the current study, although the steers that received H57-inoculated pellets had higher ruminal pH, these pH values were maintained above the critical rumen pH for fibre digestion—pH 6.0 [40]. It is speculated that an increase in ruminal pH due to H57 supplementation was possibly not sufficient to alter diet preference, and thus alteration of ruminal pH may not be an over-riding mechanism by which H57 improves preference. The advantageous effects of a probiotic supplement on ruminal pH have been shown in the studies of Nocek [41], Marden [24] and Bruno [42]. In the present study, although the rise in ruminal pH was much smaller than expected, the findings were consistent with the hypothesised ability of H57 to play a potential role in the stabilisation of rumen pH. This was supported by Le [10], who found that ruminal pH was higher for the ewes receiving H57 inoculated pellets (2.9 × 10^9^ cfu/kg pellets), than that of those who were fed with un-inoculated pellets. In contrast, Sun [43] found ruminal pH of dairy cows decreased during the supplementation of *B. subtilis natto* (0.5 × 10^11^ cfu/cow/day or at 1.0 × 10^11^ cfu /cow/day). The observation that *Bacillus* probiotic supplementation had no effect on ruminal pH was recorded by Peng [44]. These authors indicated that there was a similarity in the ruminal pH between the control cows and those supplemented with a *B. subtilis natto* fermentation product containing 8.3 × 10^9^ spores/g. The inconsistent effects of *Bacillus* probiotics on ruminal pH could be due to different strains of *Bacillus* and doses of probiotics used.

H57 supplementation increased concentrations of iso-butyrate and iso-valerate in the rumen at the 6 h time point. However, this change may not directly explain the steers’ preference for H57 during the short-term preference test. Iso-acids can regulate diet preference and DM intake of ruminants [45] because of their ability to modify the ruminal fermentation by increasing ruminal concentrations of acetate and the ratio of acetate to propionate [46,47], or enhancing the growth of important cellulolytic strains of ruminal bacteria such as *Butyrivibrio fibrisolvens*, *Ruminococcus*
*albus*, *Ruminococcus flavefaciens* and *Fibrobacter succinogenes* [48,49]. Another possible effect of iso-acids on feeding behaviour could be increased blood levels of growth hormone [47]. Although the advantages of increased concentrations of iso-acids on diet preference and DM intake were not observed in the current study, their roles in controlling feeding behaviour should be explored in further research. The increased iso-acid concentrations were in accordance with the higher levels of ruminal NH_3_ to be found in H57-fed steers. El-Shazly [50] suggested that an increase in the ruminal concentration of iso-acids was the result of the enhanced rate of protein degradation and amino acid metabolism by rumen microbes. Adding bacterial probiotics to the diet would lead to an increased iso-acids concentration, as previously recorded by Chiquette [51]. Iso-butyrate and iso-valerate productions in the rumen were increased by the probiotic *Prevotella bryantii* 25A supplemented in the dairy cow diet at the rate of 2 × 10^11^ cfu/animal/day.

Ammonia concentrations tended to be higher in ruminal samples collected in steers receiving H57 inoculated pellets orally and intra-ruminally at 2 h and 4 h. Faverdin [22] suggested that changes in the NH_3_ concentrations of the rumen could influence feeding behaviour in ruminants. While excessive amounts of NH_3_ can temporarily decrease preference through the negative actions of NH_3_ [22], a rise in ruminal NH_3_ can increase preference through their contributions to the nitrogen metabolism of the rumen microbes [52]. In this current study, a higher ruminal NH_3_ concentration could not be a cue driving the preference responses of steers for H57 pellets. This might be because the increased NH_3_ concentrations were within the range generally required for fermentation and microbial protein synthesis in the rumen [45,53], and so were not high enough to drive a change in preference. The response of ruminal NH_3_ concentration to a *Bacillus* probiotic addition varied considerably in published studies, including no change [44], increased concentrations [43] and decreased concentrations [10,54]. Although not measured directly, a higher ruminal NH_3_ concentration due to the H57 supplementation recorded in this study may be associated with higher proteolytic and deamination activities by ruminal bacteria. Schofield [29] found that feeding H57 pellets to pregnant ewes caused an increase in *Prevotella* spp. populations, which are generally major contributors to protein and peptide degradation in the rumen [55].

## 5. Conclusions

H57 increased the mature steers’ preference for H57-inoculated pellets. While it does appear that H57 supplementation can elevate ruminal pH, iso-acids and tend to increase NH_3_ concentrations, these proposed post-ingestive feedback signals were not directly linked to the steers’ preference for H57 pellets. The potential for increased ruminal iso-acids due to the H57 modulating feeding behaviour should be assessed through further experimentation. Determination of how feedback from the changes in the gut microbiota by H57 alters diet preference—potentially via microbial excretory products that can enter intermediary metabolic pathways—is required to provide more insight into the specific actions of H57 on dietary preference in ruminants.

## Figures and Tables

**Figure 1 animals-11-03455-f001:**
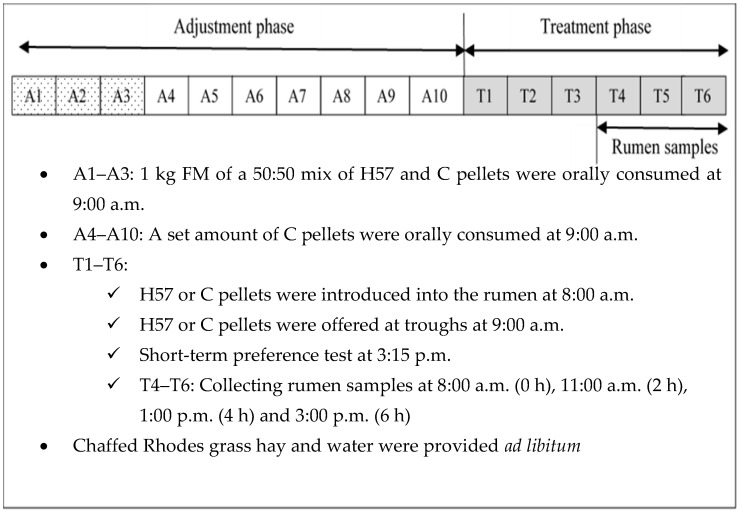
Over the course of the experiment, steers (n = 4) were subjected four times to an adjustment phase followed by a treatment phase so that they were tested on the four treatments according to either probiotic H57 (H57) or control (C) pellets that were ingested either orally (o) or introduced into the rumen (r) in the morning feeding.

**Figure 2 animals-11-03455-f002:**
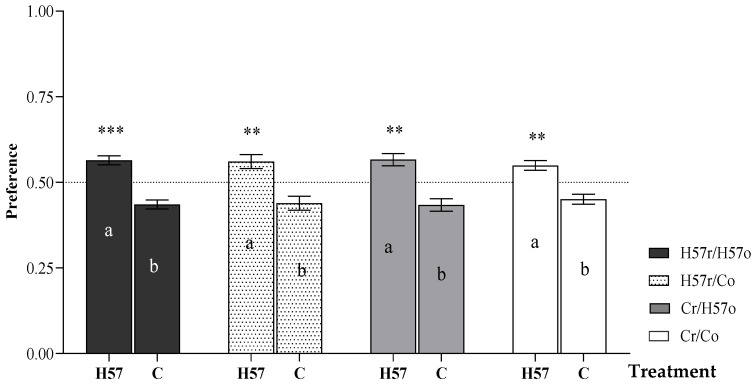
Preference of pellets during the 5 min preference test following the addition of H57 or C pellets into the digestive system, either orally (o) or through a rumen fistula (r). Asterisks indicate significant differences from the theoretical 0.5 proportion of non-preference; **: *p* < 0.01; ***: *p* < 0.001. In the columns, a, b values within the base-feeding treatment differ at *p* < 0.05; error bars represent s.e.m.

**Figure 3 animals-11-03455-f003:**
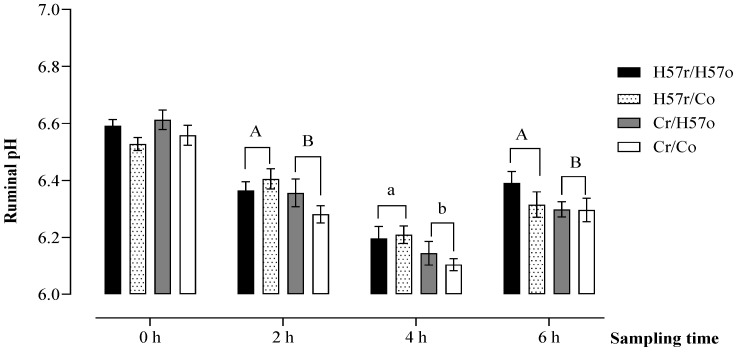
Ruminal pH following the addition of H57 or C pellets into the digestive system, either orally (o) or through a rumen fistula (r). A, B Values indicate a tendency towards statistical significance at *p* < 0.10; a, b values differ at *p* < 0.05; the *p*-value indicates the comparison between H57 (H57r/H57o and H57r/Co) and C (Cr/H57o and Cr/Co) pellets introduced directly into the rumen regardless of what was eaten from the troughs; error bars represent s.e.m.

**Figure 4 animals-11-03455-f004:**
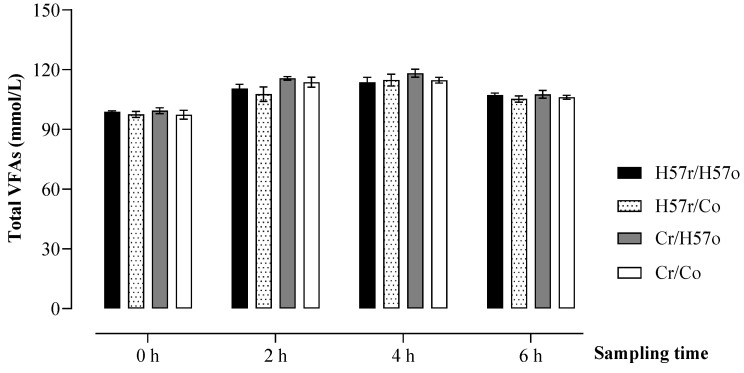
Change in ruminal volatile fatty acid (VFA) concentrations following the short-term addition of H57 or control pellets into the digestive system, either orally (o) or through a rumen fistula (r). Error bars represent s.e.m.

**Figure 5 animals-11-03455-f005:**
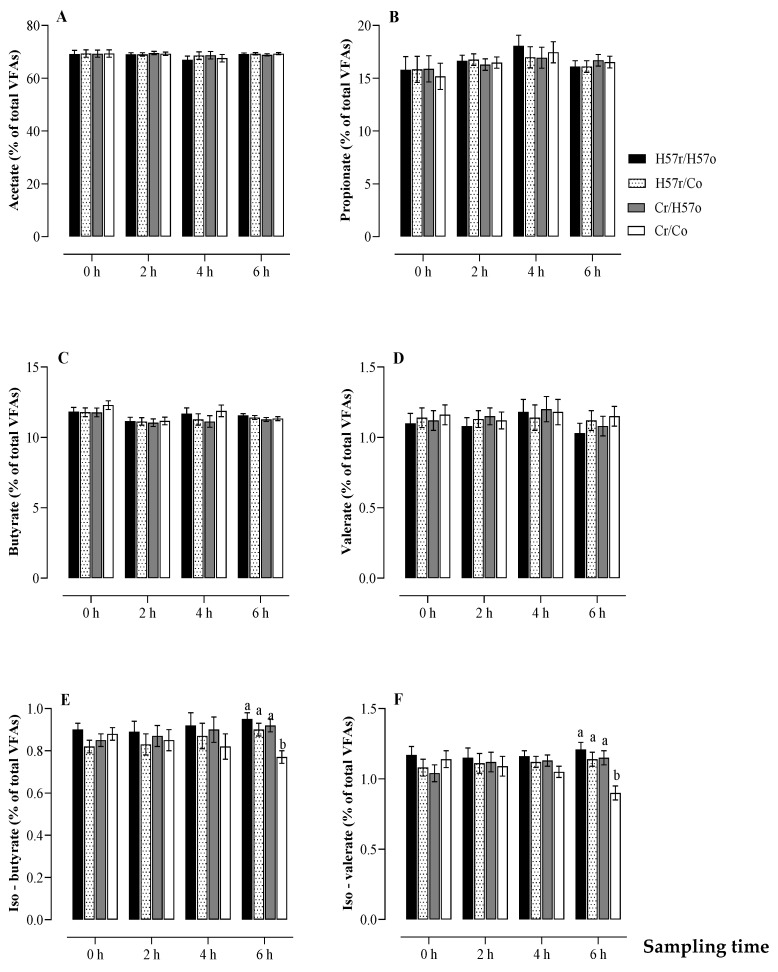
Change in ruminal molar proportions of acetate (**A**), propionate (**B**), butyrate (**C**), valerate (**D**), iso-valerate (**E**) and iso-butyrate (**F**) according to whether H57 or C pellets were ingested orally (o) or through a rumen fistula (r). a, b Values indicate statistical difference between treatments at *p* < 0.05; error bars represent s.e.m.

**Figure 6 animals-11-03455-f006:**
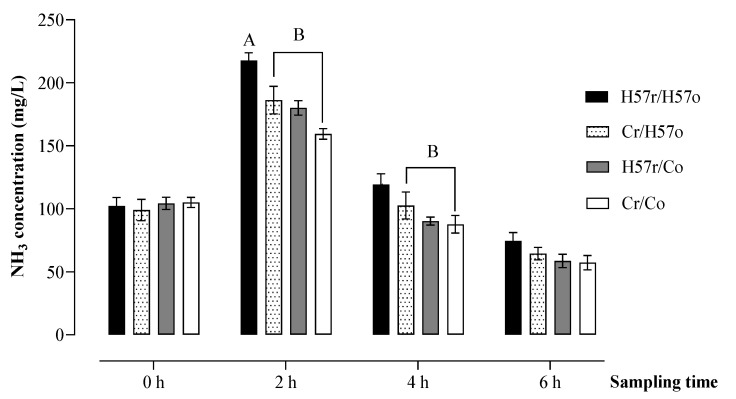
Change in ruminal ammonia (NH_3_) concentrations following the short-term addition of H57 or C pellets into the digestive system, either orally (o) or through a rumen fistula (r). A, B Values indicate a tendency towards statistical difference between treatments at *p* < 0.10; error bars represent s.e.m.

**Table 1 animals-11-03455-t001:** The ingredient and chemical components of the pellets with (H57) or without (control) the H57 and Rhodes grass hay.

	Control Pellets	H57 Pellets	Rhodes Grass Hay
Ingredients (g/kg, as-fed basis)			
Sorghum grain	494	494	-
Barley	350	350	-
Wheat	50	50	-
Cottonseed meal	45	45	-
Molasses	30	30	-
Limestone	15	15	-
Urea	5	5	-
Salt	10	10	-
Premix ^1^	1	1	-
H57 spores (cfu/g pellet, as fed)	-	1 × 10^6^	-
Chemical component (% dry matter, DM)			
Dry matter (%)	89.4	89.4	91.5
Crude protein	14.6	14.4	11.4
Fat	2.7	2.8	2.2
Acid detergent fiber	5.4	3.8	38.3
Neutral detergent fiber	9.6	8.5	57.7
Lignin	1.8	1.1	5.1
Starch	54.3	56.3	1.6
Ash	5.4	4.9	11.4
Potassium	0.54	0.56	1.75
Chloride	0.63	0.97	2.09
Sodium	0.30	0.58	1.03
Calcium	0.82	0.86	0.30
Phosphorus	0.36	0.38	0.31
Ion (ppm)	69	63	164
Zinc (ppm)	61	71	27
Metabolisable energy (MJ/kg DM)	13.4	13.8	8.7

^1^ Containing 3500 UI/g of vitamin A, 350 UI/g of vitamin D3, 30,000 UI/g of vitamin E, 40,000 mg/kg Zn, 30,000 mg/kg Mn, 10,000 mg/kg Cu, 100 mg/kg Se, 150 mg/kg Co and 300 mg/kg L.

**Table 2 animals-11-03455-t002:** Daily and short-term (5 min preference) feed intakes (kg dry matter, DM) of steers given either H57 or C pellets that were ingested either orally (o) or introduced into the rumen (r) in the morning feeding.

Voluntary Intake	Treatments				S.E.M.	*p*-Value
	H57r/H57o	H57r/Co	Cr/H57o	Cr/Co		
Intake over a 5 min preference test (kg DM/5 min)						
H57 pellets	1.84	1.78	1.77	1.73	0.07	0.74
C pellets	1.42	1.39	1.37	1.41	0.09	0.97
Total pellets	3.27	3.17	3.14	3.15	0.12	0.87
Intake over 24 h (kg DM/24 h)						
Pellets	6.49	6.38	6.41	6.40	0.12	0.90
Hay	8.22	7.78	7.82	7.52	0.32	0.52
Pellets plus hay	14.72	14.16	14.23	13.92	0.39	0.56

## Data Availability

The data presented in this study are available on request from the corresponding author. The data are not publicly available due to privacy.

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
