# Peer review of "Pellets Inoculated with Bacillus amyloliquefaciens H57 Modulates Diet Preference and Rumen Factors Associated with Appetite Regulation in Steers"

_animals, 2021, doi:10.3390/ani11123455_

Round 1

Reviewer 1 Report

The manuscript is original and the subject is interesting, but there are few variables analyzed in it.

Line 112-114: 4x4 Latin square doesn't provide sufficient degree of freedom. However, due to the animals being cannulated, this reduced number of animals can be considered.

Line 128-129: Why were pellets mixed 1:1 with artificial saliva?

Line 155: The table title should be self-explanatory. Avoid the term "pellet treatment.s" The table should have the complete ration, adding the proportion of hay, because the ration is one for all treatments. The table should also indicate which grass was used to make the hay.

Line 155-156: The equations and analyzes of "metaboliZable energy" haven't been described. It is "metabolizable energy and isn't "metabolisable energy". Cite only metabolizable energy and don't cite "total digestible nutrients".

Line 182: Is H57 facultative anaerobe?

Line 245: Avoid the term treatment after the methodology in Results, Discussion and Conclusions!

Author Response

Dear Reviewer,

Thank you for your letter and constructive comments concerning our manuscript entitled “Pellets inoculated with Bacillus amyloliquefaciens H57 modulates diet preference and rumen factors associated with appetite regulation in steers”. We have studied your comments carefully and made major corrections which we hope to meet with your approval. We answer your questions or comments in detail in the following attached file.

Kind regards,

TT Ngo

Reviewer 2 Report

Very interesting manuscript about diet preferences. Manuscript is well written and deserve to be published after minor corrections listed below:

Lines 122-124: This is not clear on Figure 1, where is shown that only C pellets were consumed. Please, make this clear on Figure 1.

Lines 130-132: Were pellets offered only for 15 min? Pellets remaining at the trough were removed after 15 min? If so, what diet did cattle consume between 9:15am 3:00pm? Please, clarify.

Line 131: Authors mentioned that time taken to eat the pellets were recorded; however, this data is not presented in the manuscript. Please, either add this data or delete this sentence.

Line 272: Figure 3, not 1.

Table 1: authors should provide the reasons by which Control and H57 present different chemical composition. H57 had 2% more starch, 1.9% more TDN, 1.1% less NDF, 1.6% less ADF, and 0.5% less ash. I do not believe that this is a sole effect of the H57 spores addition. This could explain the results of iso-acids and ruminal ammonia, since a greater amount of starch would be available for the ruminal microbes. Other question is: Were the data obtained in this study a result of H57 spores or due to different nutritional composition?

Line 283: Figure 4, not 2.

Line 364: delete “the” before “steers”.

Author Response

Dear Reviewer,

We are grateful to you for your time and constructive comments on our manuscript. We look forward to the outcome of the assessment.

Yours sincerely, 
On behalf of the co-authors 
TT Ngo

Reviewer 3 Report

The manuscript is well written and covers an interesting topic.

M&M

Table 1. How Total digestible nutrients and Metabolisable energy were calculated? And how micronutrients were determined?

L159-161. Where are those results? Are those from Table 1?

Results

Could it be possible to report differences among sampling times?

L265 – Correct Figure 3, it appears as Figure 1.

L265 -  How can you say if decreased linearly if no contrast/test was performed?

L279 – Correct Figure 2, it appears as Figure 2.

L279 – Delete dramatically

Figure 2(4). I am not fond of figures. However, it is up to the authors to use them or not. The Y-axis starts at 60, and such might cause the Figure to be misleading, as Figure might suggest differences are bigger than the really are. I am aware it was probably done to be more visually appealing, but I would recommend to modify the Figure to start at 0, considering no statistical differences were observed among treatments and p-values between sampling times are not reported.

Figure 3(as appears in the manuscript, the nh3 one) – Change to Figure 6? Correct format of the text. Same as Figure 2(4); Figure might be misleading, in both situations, it is only a trend. P-value has its advantages and problems, in this situation, considering its problems, and because it is just a trend, I would suggest to change the Figure to start at 0.

Discussion

L314- “were mildly elevated” does not seem adequate. The same words used in conclusion could be used; it does appear H57…

L360 – misleading, only a trend was observed

L382- H57 supplementation increased concentrations… after 6 hours.

L400 – Ammonia concentrations tended … after 2 and 4 hours

General comments

Check P-values format in the text.

Check keywords and use.

Author Response

Dear Reviewer,

Thank you for reading our manuscript and reviewing it, which will help us improve it to a better scientific level. We revised our manuscript, and changes have taken place. Thus, we have sent the revised manuscript, and a version containing all the changes, which we hope to meet with your approval.

Kind regards,

TT Ngo

Round 2

Reviewer 1 Report

The authors made all corrections, but I think that are little results to be publish in Animals. I'd like that the authors should add more results in manuscript.

Author Response

Dear Reviewer,

We thank you for your time spent carefully reviewing the manuscript, and for your opinions regarding the science and presentation of the material.

Below, we provide a point-by-point response explaining how we have addressed the reviewer’s comments. We look forward to the outcome of your assessment

Yours sincerely,

On behalf of the co-authors

TT Ngo
